**Data Availability Statement:** All relevant data are within the manuscript and its Supporting Information files.

# Pre-lacteal feeding practices and associated factors among mothers of children aged less than 12 months in Jinka Town, South Ethiopia, 2018/19

**Muluken Bekele Sorrie**[1]*, **Elias Amaje**[2], **Feleke Gebremeskel**[1]

**1** School of Public Health, College of Medicine and Health Sciences, Arba Minch University, Arba Minch, Ethiopia, **2** Department of Public Health, Bule Hora University, Hagere Maryam, Ethiopia

* mulukenbekele51@gmail.com

## Abstract

### Background

Pre-lacteal feeding is one of the major harmful practices being faced while feeding the newborns. Although it affects child health, little is known about the extent of the problem and its contributing factors in the study area. Therefore, this study aimed to figure the prevalence of pre-lacteal feeding practices and associated factors among mothers of children aged less than 12 months in Jinka Town.

### Methods

A community-based cross-sectional study was conducted at Jinka Town from March 1 to 30, 2019. A total of 430 mothers, having children less than 12 months of age, were selected by systematic sampling technique. The data were collected by using pretested and interviewer-administered structured questionnaires. The data were entered using epidata 4.2.1 and exported to SPSS version 23 for analysis. Adjusted odds ratios, 95% confidence intervals and p-values were reported.

### Results

The prevalence of pre-lacteal feeding practice was 12.6% [95% CI (9.5–15.7)]. Having no maternal education [AOR = 4.82(95%CI 1.60–14.24)], colostrum avoidance [AOR = 4.09 (95% CI 1.62–7.67)], lack of breast feeding counseling [AOR: = 2.51(95% CI 1.20–5.25)], home delivery [AOR = 3.34 (95% CI 1.52–7.33)], lack of knowledge about risks of pre-lacteal feeding [AOR = 2.86 (95% CI 1.30–6.29] and poor knowledge on breast feeding practices [AOR = 3.63(95% CI 1.62–8.11)] were factors associated with pre-lacteal feeding practices.

### Conclusion

Pre-lacteal feeding practice among mothers of children aged less than 12 months in Jinka town was found to be higher than the national prevalence. Illiterate, colostrum avoidance,

**Funding:** The author(s) received no specific funding for this work.

**Competing interests:** The authors have declared that no competing interests exist

**Abbreviations:** ANC, Antenatal Care; AOR, Adjusted Odds Ratio; CI, Confidence Interval; COR, Crude Odds Ratio; CSA, Central Statistical Agency; EBF, Exclusive Breast Feeding; EDHS, Ethiopian Demographic and Health Survey; IYCF, Infant and Young Feeding Practice; PLF, Pre-lacteal Feeding; SPSS, Statistical Package for Social Sciences; UNICEF, united Nations children's emergency fund; WHO, World Health Organization.

lack of breastfeeding counseling, home delivery, lack of knowledge on the risk of pre-lacteal feeding, and poor knowledge on breastfeeding practice were factors associated with pre-lacteal feeding practices.

## Introduction

World Health Organization(WHO) recommends, exclusive breastfeeding up to 6 months of age, continued breastfeeding along with right complementary foods up to two years of age or beyond by emphasizing the newborn should start breastfeeding within an hour after birth [1]. It provides immense immunological, psychological, socio-economic, and environmental benefits [2]. It also significantly reduces a child's risk of developing obesity, type 2 diabetes mellitus, and related chronic non-communicable diseases [3]. However, in different countries including Ethiopia, significant proportions of mothers offer pre-lacteal foods to their newborns [4–6]. Pre-lacteal foods are any food given to newborns before breastfeeding is started in the first 3 days of life [7]. Clean water, rice water, herbal mixture, and milk based foods are the most common pre-lacteal foods given to newborns in low and middle-income countries [4]. Even if immediate and exclusive breastfeeding helps to support healthy growth in infants and protection against infections [8], pre-lacteal foods interfere with the development of an ideal gut microbe leading to infantile diarrhea, which in turn affects newborn's development [9]. it also disrupts the establishment of normal flora in the gastrointestinal tract [5].

For example, indicate studies in different parts of the world revealed that pre-lacteal feeding is a prevailing problem. A report from Vietnam, India, and Nepal showed a high prevalence of pre-lacteal feeding, 73.3%, 49.5%, and 30.6% respectively [4, 5, 10]. In Sub-Sahara African countries, the prevalence of pre-lacteal feeding was 60.5% in Nigeria, 36.1% in Burkina Faso, 15.5% in Kenya, and 15% in Ghana [11]. Findings from different parts of Ethiopia showed that pre-lacteal feeding is the common nutritional malpractice as 42.9% of mothers in Afar, 45.4% in Harar, 38.8% in Raya Kobo, and 10.1% in Axum reported that they gave pre-lacteal foods to their infant [6, 12–14].

Due to the introduction of pre-lacteal foods, 3000–4000 infants die every day in the developing world from diarrhea and acute respiratory infections [15]. In particular, 45% of neonatal infectious deaths, 30% of diarrheal deaths, and 18% of acute respiratory deaths among under-five children were associated with pre-lacteal feeds [16, 17].

Pre-lacteal food has a great impact on the newborns' mental health, physical development, and fighting against infections [18]. Besides, this feeding process reduces the practice of exclusive breastfeeding, which can be dangerous to the child and results in early cessation of breastfeeding [19]. It is also associated with a more likely chance of dying in the neonatal period [12].

Different factors affect pre-lacteal feeding practice, mainly related to home delivery, failure to attend ANC, late breastfeeding initiation, and influence by friends [10]. Birth order of index child, birth spacing less than 24 months, colostrum discarding, delivery by cesarean section and maternal belief on the purported advantage of pre-lacteal feeding were also factors affecting pre-lacteal feeding [14]. Pre-lacteal feeding was also highly affected by maternal educational status, giving birth to a male and previous experience on pre-lacteal feeding [4].

A wide range of harmful newborn feeding practices are documented in Ethiopia after the implementation of infant and young child feeding guidelines [20]. Knowing the severity as well as the wide-spreading practices of inappropriate breastfeeding, the government of

Ethiopia has devised different strategies including generating health extension programs and working in collaboration with Non-Governmental Organizations (NGOs) in the areas of IYCF [21, 22]. WHO and UNICEF recommend that children should start breastfeeding within the first hour of birth and no other foods or liquids should be provided, including water before the starting of breastfeeding, but in Ethiopia, about 7.9% of children received pre-lacteal feeding [7, 23].

Consumption of animal products is high in pastoral areas where cows or goat milk is a major part of the diet for pastoralist children in addition to breastfeeding and mothers living in the area tend to give this for their infants before 6 months due to low educational level and awareness status.

Although pre-lacteal feeding affects child health, little is known about the extent of the problem and its contributing factors in South Ethiopia particularly in the study area. Even studies conducted on this and related topics show inconsistencies among their findings. Therefore, the purpose of this study is to assess the prevalence of pre-lacteal feeding practices and associated factors among mothers of children aged less than 12 months in Jinka Town, Southern Ethiopia.

## Methods

### Setting, design and period of the study

This study was conducted in Jinka Town, South Omo Zone, and located 755 Km away from Addis Ababa and 525 Km from Hawassa. The town has an estimated population size of 31,226 living in 6 kebeles (the smallest administration unit in Ethiopia). Out of the total population, 15,582 are men and 15,644 are women. Out of the female population, 6,076 are women in the reproductive age group (15–49 year). About 997 of the total population are children less than one year of age. The Town has 1 hospital, 1 health center, and 6 health posts providing health services including maternal and child health care. The town also has 12 private clinics and 13 drug vendors [24].

A community-based cross-sectional study was conducted to assess the prevalence of pre-lacteal feeding practice and associated factors among mothers of children aged less than 12 months in Jinka Town, South Omo Zone, Ethiopia from March 1 to 30, 2019.

### Population of the study

All mothers of children aged less than 12 months in Jinka Town South Omo Zone was the source population where all mothers of children aged less than 12 months in the selected kebeles of Jinka Town during data collection period were the study population.

### Eligibility criteria

#### Inclusion criteria

All mothers/caregivers of children aged less than 12 months and mothers who had lived for at least 6 months in the study area were included in this study, while those mothers who were seriously ill or unable to give the required information during the data collection period were excluded.

#### Sample size determination and sampling technique

In this study, the sample size is determined by using a single population proportion formula. Considering the prevalence of pre-lacteal feeding practice of 20.3% obtained from the previous study conducted in Motta Town [25], assuming a 95% confidence level, 5% of margin of error,

the design effect of 1.5 and by adding 15% of the non-response rate, the total sample size was 430.

We used a systematic sampling technique to select study participants. From a total of 6 kebeles of Jinka Town 4 Kebele's were selected by Lottery Method. To get the sample size from each selected kebele's, proportional allocation to sample size was done. First, the numbering of all households of selected kebele with mothers of children aged less than 12 months was conducted, and then a systematic sampling technique was applied for the selection of study participants. Finally, every K value of 2 mothers from each household of the selected kebele was identified until the required sample size fulfilled and the starting household was selected using a lottery method. At the time of the survey, from each household unit, one eligible mother who had a child aged less than 12 months were selected. When there was more than one potential respondent in a household, simple random sampling was done to select one.

## Operational definition and definition of terms

### Pre-lacteal feeding

If an infant within the first three days of life feeds something other than breast milk. Accordingly, a mother was asked a key question to find out pre-lacteal feeding practice. The mother was asked if gave any drink other than breast milk to the child within the first three days of delivery. If she responded "yes" it was coded '1', otherwise coded '0' as she didn't give any pre-lacteal feed [7].

### Delayed initiation of BF

Initiation of breastfeeding after one hour of birth [7].

### Good knowledge of breastfeeding practice

If a mother answered four questions out of seven on breastfeeding knowledge correctly [25].

## Data collection tools and procedure

Data was collected by using a structured interviewer-administered questionnaire. Data were collected by four trained diploma nurses who are fluent speakers in the local language and supervised by two BSc public health professionals. The questionnaire was constructed by adapting from previous literature [4, 5, 13, 14, 25] and contextualized to fit the research objective. The questionnaire mainly addressed socio-demographic, infant feeding, maternal health services utilization, maternal health related, misconception related, and maternal knowledge on breastfeeding practices.

## Data quality control

To assure the quality of data, properly designed data collection instruments were provided after proper training for data collectors and supervisors. The questionnaire was initially prepared in English and then translated into Amharic version (local language) by different fluent speakers of both languages and then to English to check its consistency. The questionnaire was pretested before the real data collection on 5% of the sample size in nearby Town, Key Afer to check clarity and consistency of data collection instruments. During pre-testing, an effort was made to check for consistency in the interpretation of data collection tools and to find ambiguous items. The collected data were checked for consistency, completeness, and relevance daily during the entire data collection period by the supervisors and principal investigator.

## Data processing and analysis

The collected data was coded and entered by epidata 4.4.2.1 and exported to statistical package for social science (SPSS) version 23.0 for analysis. Then data cleaning, editing, and management were carried out. The household wealth index was computed by considering properties, like selected household assets. The wealth index of participants' households was computed by the principal component (PCA). Binary logistic regression analysis was employed to check the statistical association between PLF practices and independent variables. Variables that have p-value < 0.25 during bivariate analysis were entered into a multivariable logistic regression to find statistically significant variables. The model goodness of fit was tested by Hosmer-Lemeshow statistic which is not significant p-value = 0.151. Multi colinearity test was carried out to see the correlation between all independent variables using collinearity statistics which is tolerance > 0.1, variance inflation factor < 10 and standard error which was less than 4. An Adjusted odds ratio (AOR) with 95% CI at a p-value < 0.05 was estimated to find statistically significant variables. The result was presented using tables and text.

## Ethics approval and consent to participate

The study protocol was approved by the Institutional Board (IRB) of the College of Medicine and Health Science, Arba Minch University. Based on the approval, an official letter of support was written by AMU Public Health Department to Jinka Town health office. The aim of the study was explained and verbal consent (the consent form was read to the participants and when they declare their voluntary consent to take part in the study verbally) was secured from the study participants. The right of participants to withdraw from the study at any time without any precondition was disclosed. Moreover, the confidentiality of the information obtained was guaranteed by all data collectors and investigators.

## Results

### Socio demographic characteristics of mothers and children

Four hundred twenty mothers having children less than 12 months of age were interviewed in this study, with a response rate of 97.7%. The mean age of respondents was 26.86 ±5.06 years SD. The majority of the respondents; 314(74.8%) were unemployed by occupation. Around half of the children, 228 (54.3%) were males (Table 1).

### Prevalence of pre-lacteal feeding practices

The prevalence of pre-lacteal feeding practice in this study was 53(12.6%) with 95% CI [9.5–15.7]. This implies that 12.6% of study participants were reported that they have given pre-lacteal foods to their newborn in the first three days of birth. The most common type of pre-lacteal food was plain water 22(41.5%) followed by butter 13 (24.5%), cow milk (7.5%), and glucose water (5.7%).

### Decisions and reasons for pre-lacteal feeding practices

The majority of the respondents from those who practice pre-lacteal feeding (41.5%) gave pre lacteal feeding for their newborns with their own decision, and (22.6%) of respondents provide PLF due to grandparents' advice. Three hundred seventy-nine (90.2%) of respondents fed colostrum for their infants within the first five days after delivery and 41(9.8%) of respondents avoided colostrum. The main reasons for colostrum avoidance were breast milk insufficiency 14(34.1%) followed by considering colostrum causes abdominal discomfort and diarrhea for the newborn 11(26.8%). Three hundred twenty-four (77.1%) of mothers were initiated

**Table 1. Socio-demographic characteristics of mothers of children aged less than 12 in Jinka Town, South Ethiopia, 2018/19 (N = 420).**

| Variables | Category | Frequency | Percentage |
|---|---|---|---|
| Sex of the index child | Male | 228 | 54.3 |
| | Female | 192 | 45.7 |
| Maternal age | 15–24 | 141 | 33.5 |
| | 25–34 | 235 | 56.0 |
| | ≥35 | 44 | 10.5 |
| Marital status | Married | 396 | 94.3 |
| | Unmarried | 24 | 5.7 |
| Educational status of mother | Unable to read and write | 63 | 15.0 |
| | Primary education | 183 | 43.6 |
| | Secondary and above | 174 | 41.4 |
| Religion | Orthodox | 259 | 61.7 |
| | Protestant | 139 | 33.1 |
| | Muslim | 19 | 4.5 |
| | Others | 13 | 0.7 |
| Ethnicity | Amhara | 181 | 43.1 |
| | Ari | 114 | 27.1 |
| | Wolayta | 42 | 10.0 |
| | Basketo | 42 | 10.0 |
| | Gofa | 31 | 7.4 |
| | Others | 10 | 2.4 |
| Maternal occupation | Unemployed | 314 | 74.8 |
| | Employed | 106 | 25.2 |
| Educational status of the father of index child | Unable to read and write | 40 | 9.5 |
| | Primary education | 133 | 31.7 |
| | Secondary and above | 247 | 58.8 |
| Family size | ≥4 | 280 | 66.7 |
| | ≤3 | 140 | 33.3 |
| Wealth index | Poor | 140 | 33.3 |
| | Middle | 146 | 34.8 |
| | Rich | 134 | 31.9 |

N = 420

breastfeeding within one hour, while the remaining 96(22.9%) initiated breastfeeding for more than one hour.

## Maternal health care service utilization and obstetric characteristics

Three hundred eighty (90.5%) of mothers have used ANC services for their index infants. From those mothers who have used ANC services 140(36.8%) were used four times and above. About 268(63.8%) of respondents have got breastfeeding counseling. From those mothers who were counseled on breastfeeding 154(57.5%) counseled on the benefits of breastfeeding (Table 2).

## Maternal medical condition and breast problem

The majority (89.3%) of mothers did not face any of the breast problems after the delivery of the index child. From breast problem, mastitis (35.8%) was the most common problem

**Table 2. Maternal health care service utilization among mothers of children aged less than 12 months in Jinka Town, South Ethiopia, 2018/19 (N = 420).**

| Variables | Category | Frequency | Percentage |
|---|---|---|---|
| Attending ANC (N = 420) | Yes | 380 | 90.5 |
| | No | 40 | 9.5 |
| Number of Antenatal visit (N = 387) | 1–3 visit | 240 | 63.2 |
| | ≥4 visit | 140 | 36.8 |
| Breastfeeding counseling | Yes | 268 | 63.8 |
| | No | 152 | 36.2 |
| Place of delivery (N = 420) | Health facility | 327 | 77.9 |
| | Home | 93 | 22.1 |
| Mode of delivery (N = 420) | Spontaneous delivery | 351 | 83.6 |
| | Instrumental delivery | 28 | 6.7 |
| | C/S delivery | 41 | 9.8 |
| PNC | Yes | 273 | 65.0 |
| | No | 147 | 35.0 |
| Number of PNC visit | 1 | 64 | 23.4 |
| | 2 | 131 | 48.0 |
| | 3 | 78 | 28.6 |
| Birth order of index child | 1 | 148 | 35.2 |
| | 2–3 | 198 | 47.2 |
| | ≥4 | 74 | 17.6 |
| Birth spacing | No previous child | 148 | 35.2 |
| | <24 months | 64 | 15.3 |
| | ≥24 months | 208 | 49.5 |

N = 420

mothers faced followed by breast milk insufficiency (31.1%), abscess (17.8%) and cracked/sore nipples (15.6%). Majority 379(90.2%) of the mothers not faced any medical illness after the delivery of index child.

## Maternal belief on advantages of PLF

In this study, 355 (84.5%) mothers did not report their belief on advantages of pre-lacteal feeding for infants while the remaining reported. Twenty-nine mothers who believe and the advantages of PLF were reported that PLF was important for child health whereas 25 mothers reported its importance to clean infants' bowel. Regarding the previous experience of PLF, 361 (86%) of respondents were reported that they had no previous experiences of PLF.

## Maternal knowledge on risks of PLF

Two hundred fifty-six (61.0%) of respondents were stated that they knew the risk associated with pre-lacteal feeding. The majority of mothers were reported that infection 109(42.2%), diarrhea 99(38.4%), vomiting 43(16.7%), and poor growth 41 (15.9%) were the main problem related to pre-lacteal feeding.

## Maternal knowledge on breastfeeding

From total respondents, 290 (69%) of mothers knew about as there is no need to give pre-lacteal feeding to the infant and 361(86%) of mothers knew about the importance of colostrum for the infant. Three hundred fifty-two (83.8%) mothers had good knowledge about optimal

**Table 3. Breastfeeding knowledge of mothers of children aged less than 12 months in Jinka Town, South Ethiopia, 2018/19 (n = 420).**

| Knowledge questions | Response | Frequency | Percentage |
|---|---|---|---|
| Breastfeeding is important for infant health | True | 400 | 95.2 |
|  | False | 20 | 4.8 |
| Breastfeeding is important for maternal health | True | 271 | 64.5 |
|  | False | 149 | 35.5 |
| An infant should be put to breast immediately after birth | True | 346 | 82.4 |
|  | False | 74 | 17.6 |
| The first milk/colostrum should be given to an infant | True | 361 | 86.0 |
|  | False | 59 | 14.0 |
| Pre-lacteal feeding is not needed for an infant before starting breast milk | True | 290 | 69.0 |
|  | False | 130 | 31.0 |
| Breast milk alone without water and other liquids is enough for an infant during the first 6 months of life | True | 274 | 65.2 |
|  | False | 146 | 34.8 |
| Starting from 6 month an infant should start complementary feeding and continued breastfeeding up to 2 years and beyond | True | 289 | 68.8 |
|  | False | 131 | 31.2 |

N = 420

breastfeeding practice while the remaining 68(16.2%) had poor knowledge about optimal breastfeeding practices (Table 3).

## Factors associated with pre-lacteal feeding practices

In the last multivariable logistic regression analysis, the educational status of the mother, colostrum avoidance, BF counseling, place of delivery, knowledge on risks of PLF, and knowledge on BF practices were factors associated with PLF practices. The Mothers who were unable to read and write were 4.82 times more likely to give pre-lacteal foods when compared to mothers with secondary education and above (AOR = 4.82, 95% CI; 1.6–14.24). Colostrum avoiding mothers were 4 times more likely to give PLF to their newborns in comparison with their counterparts (AOR = 4.09, 95% CI; 1.26–13.25). The Mothers who didn't get breastfeeding counseling was 2.51 times more likely to give PLF as compared to their counterparts' (AOR = 2.51, 95% CI; 1.20–5.25). Mothers who delivered the index child at home were 3.34 times more likely to practice PLF compared to their counterparts (AOR = 3.34, 95% CI; 1.52–7.33). Mothers who didn't know the risks of PLF were 2.86 times more likely to give PLF when compared to their counterparts (AOR = 2.86,95% CI; 1.30–6.29). Furthermore, PLF practice was 3.63 times higher among mothers with poor knowledge of BF practices when compared to their counterparts (AOR = 3.63, 95% CI; 1.62–8.11) (Table 4).

## Discussion

The prevalence of pre-lacteal feeding practice in Jinka town was 12.6%. This makes breast feeding practices sub-optimal in Jinka town due to the introduction of pre-lacteal feeding to the newborns. This finding is similar to the study done in Axum town 10.1%, Mettu district 14.2%, North Eastern Ethiopia 11.1%, and Debrebirhan district 14.2% respectively [14, 26–28]. The finding of this study is also consistent with the study conducted in Tamilnadu, India14.8%, Post-conflict Timor-Leste 12.3%, and Benin, Nigeria 11.7% respectively [29–31]. However, the finding of this study was higher than the 2016 Ethiopian DHS report 7.9%, the studies done in East Wollega, West Ethiopia 6.7% and Offa district, southern Ethiopia 6.1% respectively [7, 32, 33]. The difference between these studies might be due to the difference in

**Table 4. Bivariable and multivariable logistic regression analysis of factors associated with pre-lacteal feeding practices among mothers of children aged less than 12 months in Jinka Town, 2018/19.**

| Variables | Categories | Pre-lacteal feeding practice | | Crude OR(95% CI) | Adjusted OR(95% CI) |
|---|---|---|---|---|---|
| | | Yes (%) | No (%) | | |
| Educational status of Mother | Unable to read and write | 22(41.5) | 41(11.2) | 9.83(4.21–22.9) | **4.82(1.6–14.24)** * |
| | Primary education | 22(41.5) | 161(43.9) | 2.50(1.12–5.60) | 1.47(0.58–3.72) |
| | Secondary and above | 9(17) | 165(44.9) | 1 | 1 |
| Maternal occupation | Unemployed | 46(86.8) | 268(73.1) | 1 | |
| | Employed | 7(13.2) | 99(26.9) | 0.41(0.18–0.94) | 0.53(0.14–1.88) |
| Sex of child | Male | 35(66.1) | 193(52.6) | 1.75(0.95–3.20) | 2.12(0.97–4.62) |
| | Female | 18(33.9) | 174(47.4) | 1 | |
| Colostrum feeding | Yes | 38(71.7) | 341(92.9) | 1 | 1 |
| | No | 15(28.3) | 26(7.1) | 5.17(2.5–10.6) | **4.09(1.26–13.2)** * |
| BF initiation | Timely | 26(49.1) | 298(80.8) | 1 | 1 |
| | Delayed | 27(51.9) | 69(19.2) | 4.48(2.46–8.16) | 2.20(0.92–5.24) |
| BF counseling | Yes | 19(35.8) | 249(67.8) | 1 | 1 |
| | No | 34(64.2) | 118(32.2) | 3.77(2.06–6.89) | **2.51(1.20–5.25)** * |
| Place of delivery | Health facility | 22(41.5) | 305(83.1) | 1 | 1 |
| | Home | 31(58.5) | 62(16.9) | 6.93(3.76–12.7) | **3.34(1.52–7.33)** * |
| Maternal belief on PLF | Yes | 19(35.8) | 46(12.5) | 3.90(2.05–7.40) | 1.50(0.61–3.65) |
| | No | 34(64.2) | 321(87.5) | 1 | |
| Previous experience of PLF | Yes | 17(32.1) | 42(11.4) | 3.65(1.88–7.02) | 2.55(0.99–6.55) |
| | No | 36(67.9) | 325(88.6) | 1 | |
| Knowledge on risks of PLF | Yes | 15(28.3) | 241(65.7) | 1 | 1 |
| | No | 38(71.7) | 126(34.3) | 4.84(2.56–9.14) | **2.86(1.30–6.29)** * |
| Knowledge on BF practice | Good | 28(52.8) | 332(88.3) | 1 | 1 |
| | Poor | 25(47.2) | 35(11.7) | 8.46(4.45–16.1) | **3.63(1.62–8.11)** * |

Key:

* = statistically significant at p<0.05 in multivariable logistic regression

1 = the reference category

community attitude towards PLF among ethnic groups. The other possible reason for this inconsistency might be the socio-demographic difference among study participants.

The finding of this study is also lower than studies done in different corners of Ethiopia. Raya Kobo district 38.8%, Eastern Ethiopia, which was 45.4%, Afar region 42.9%, Dabat district 26.8%, Debre Markos Town 19.1% and Motta Town 20.3% [6, 12, 13, 25, 34, 35]. The possible reason for this difference might be the difference in age of the child between this study and above studies at which the majority of the above studies were carried out among mothers of children aged less than 24 months where mothers may face difficulties to remember what they fed their child. Also, a study conducted in eastern Ethiopia was facility-based it is assumed that mothers with good educational status have a high chance of visiting health centers. The finding of this study is also lower than studies carried out in Nepal 26.5%, Vietnam 73.3%, Karnakata, India 32.03%, Egypt 58%, Kampala, Uganda 31.3%, and South Sudan 53% [4, 5, 19, 36–38]. This difference could be due to the difference in maternal health service utilization between study populations. The other possible reasons could be due to the difference in year of the study and age of the child between this study and above studies at which the majority of the previous studies were done among mothers of children aged less than 6 months.

The odds of providing pre-lacteal feeding among mothers who were unable to read and write were nearly five folds higher than those who were secondary and above educational level. This finding was supported by evidence of studies in Nepal, Debre Markos Town, and Mettu District [4, 28, 34]. The possible reason might be maternal education increases mothers' level of awareness about the importance of right breastfeeding that makes mothers not introduce pre-lacteal feeds to their newborns. Mothers with little or no education might be more likely to be influenced by traditional birth attendants and grandparents that can influence mothers to practice PLF. The findings from a study conducted in Mansour, Egypt contradicts this idea showed that mothers who were highly educated were 1.7 times more likely to give pre-lacteal feeds for their newborns when compared to the mother who was secondary and below in educational status [38].

In this study, mothers who discarded colostrum in the first 5 days were about four times more likely to practice pre-lacteal feeding than those who give colostrum to their index child. This result is consistent with the study done in Axum town, Mettu district, Motta town, and North Eastern Ethiopia, respectively [14, 25, 27, 28]. This might be because when the mother avoids colostrum infant suckling activity decreases and which in turn affects or decreases maternal milk secretion due to decreased breast stimulation, which finally made the mother give other food to the infant [25]. This might be due to the mother's belief in considering colostrum as unclean and bad for the infant's health.

This study showed that mothers who didn't get breastfeeding counseling were 2.5 times more likely to practice PLF when compared to their counterparts. Similar findings were reported from Vietnam, South Sudan, and North Eastern Ethiopia [5, 27, 36]. This might be counseling is the tool to change the behaviors of mothers to reduce pre-lacteal feeding practice during the time of pregnancy. This could be breastfeeding counseling during the prenatal period may increase the mother's awareness of optimal breastfeeding practices that might decrease PLF practices. However, the study conducted in Axum town and Debre Markos town revealed that there was no association between breastfeeding counseling during ANC visit and PLF practices [14, 34].

In this study, mothers who delivered their index infants at home were 3.34 times more likely to engage in pre-lacteal feeding practices when compared with those who delivered in the health facility. This finding was consistent with a study conducted in Raya Kobo district, Harar region, Mettu district, and Debrebirhan district, Ethiopia [6, 13, 26, 28]. This indicates that strengthening maternal health service improves optimal breastfeeding practices. This could be because mothers, who gave birth at home, were more likely to be exposed to the traditional beliefs that favor pre-lacteal feeding like the child will not gain adequate water, important to clean infants' throat/bowel. In contrast, utilizing an institutional delivery would have an added benefit to receiving immediate obstetrical care, such as early initiation of breastfeeding which reduces the likelihood of giving pre-lacteal feeding [6]. This finding was contrary to the study done in South Sudan and North East Ethiopia that revealed the place of delivery was not associated with PLF practices [27, 36].

In this study, mothers who did not know the risks of PLF were 2.86 times more likely to practice PLF when compared to their counterparts. This finding was similar to studies done in Raya Kobo district, Mettu district, Ethiopia [6, 28]. The possible justification might be that if mothers did not know risks associated with PLF, the influence of local community members especially the grandparents and traditional birth attendants might stress them to give pre-lacteal feeding.

The odds of PLF were 3.63 times higher among mothers with poor knowledge of optimal breastfeeding practices when compared to their counterparts. This finding was supported by other studies carried out in Vietnam, Afar Region, and Dabat district [5, 12, 35]. This

supporting evidence revealed that improving the mother's awareness of optimal infant feeding practices reduces the likelihood of PLF. This might be due to the awareness of mothers about breastfeeding practices and the nutritional value of colostrum decreases the likelihood of practicing PLF. However, the study done in Eastern Ethiopia showed that maternal knowledge of optimal breastfeeding practice was not associated with PLF [13].

## Limitation of the study

The limitation of this study was that information obtained from mothers having children aged less than 12 months are subject to recall bias. The study also shares the limitation of the cross-sectional study design. It may not be representative of the nation

## Conclusion and recommendation

Pre-lacteal feeding practice among mothers of children aged less than 12 months in Jinka town was found to be higher than the national prevalence. Independent factors associated with PLF practices are being unable to read and write, colostrum avoidance, lack of breast-feeding counseling, home delivery, lack of knowledge on the risk of PLF and poor knowledge on breastfeeding feeding practices.

The recommended interventions to reduce PLF practices in Jinka town are awareness creation activities on the risks of PLF, promotion of institutional delivery and improving breastfeeding counseling. Interventions to reduce PLF should also target grandparents, and traditional birth attendants within the study area.

## Supporting information

**S1 Data.**
(SAV)

**S1 Questionnaire.**
(DOCX)

## Acknowledgments

We would like to express our heartfelt appreciation to the data collectors, supervisors and study participants. We would like to express our appreciation for Mr. Zewde Jagre and Mr. Endelibu Gao for language editing of the article.

## Author Contributions

**Conceptualization:** Elias Amaje.

**Formal analysis:** Muluken Bekele Sorrie, Elias Amaje.

**Methodology:** Muluken Bekele Sorrie, Elias Amaje, Feleke Gebremeskel.

**Supervision:** Muluken Bekele Sorrie, Feleke Gebremeskel.

**Writing – original draft:** Elias Amaje, Feleke Gebremeskel.

**Writing – review & editing:** Muluken Bekele Sorrie, Elias Amaje, Feleke Gebremeskel.

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
