## [Decision Letter · Decision Letter 0]

5 Feb 2020

PONE-D-19-26600

Pre-lacteal Feeding Practices and Associated Factors among Mothers of Children Aged Less Than 12 Months in Jinka Town, South Ethiopia, 2018/19

PLOS ONE

Dear Mr Sorrie,

Thank you for submitting your manuscript to PLOS ONE. After careful consideration, we feel that it has merit but does not fully meet PLOS ONE’s publication criteria as it currently stands. Therefore, we invite you to submit a revised version of the manuscript that addresses the points raised during the review process.

The manuscript has been evaluated by three reviewers, and their comments are available below.

<h2>**The reviewers have raised a number of concerns that need attention. They request additional discussion of the potential limitations of this study, including the lack of generalizability of the results. All reviewers also mention that the manuscript requires copyediting in order to meet PLOS ONE publication criterion #5 which states that “The article is presented in an intelligible fashion and is written in standard English.”.**</h2>

Could you please revise the manuscript to carefully address the concerns raised?

We would appreciate receiving your revised manuscript by Mar 20 2020 11:59PM. To enhance the reproducibility of your results, we recommend that if applicable you deposit your laboratory protocols in protocols.io, where a protocol can be assigned its own identifier (DOI) such that it can be cited independently in the future. For instructions see: http://journals.plos.org/plosone/s/submission-guidelines#loc-laboratory-protocols

We look forward to receiving your revised manuscript.

Kind regards,

Natasha Rickett

Academic Editor

PLOS ONE

Journal Requirements:

5. Your ethics statement must appear in the Methods section of your manuscript. If your ethics statement is written in any section besides the Methods, please move it to the Methods section and delete it from any other section. Please also ensure that your ethics statement is included in your manuscript, as the ethics section of your online submission will not be published alongside your manuscript.

Reviewers' comments:

Reviewer's Responses to Questions

**Comments to the Author**

1. Is the manuscript technically sound, and do the data support the conclusions?

Reviewer #1: Yes

Reviewer #2: Yes

Reviewer #3: Yes

2. Has the statistical analysis been performed appropriately and rigorously? 

Reviewer #1: Yes

Reviewer #2: I Don't Know

Reviewer #3: Yes

3. Have the authors made all data underlying the findings in their manuscript fully available?

Reviewer #1: Yes

Reviewer #2: Yes

Reviewer #3: No

4. Is the manuscript presented in an intelligible fashion and written in standard English?

Reviewer #1: Yes

Reviewer #2: No

Reviewer #3: No

5. Review Comments to the Author

Reviewer #1: Dear Authors,

The article titled as “Pre-lacteal Feeding Practices and Associated Factors among Mothers of Children Aged Less Than 12 Months in Jinka Town, South Ethiopia, 2018/19” was submitted by the research team in Ethiopia.

The study was not published in any other journals.

Pre-lacteal feeding is one of the major harmful newborn feeding practice in developing countries. The recommendation of WHO is “exclusive breastfeeding up to 6 months”, it provides health growth in the first months of the life. The topic of the study is very important to evaluate the child health status in the field.

The study is cross-sectional, and representative for Jinka Town, South Omo Zone, Ethiopia in March 2019. Sample size was determined by using a single population proportion formula and 4 out of 6 Kebele were selected by Lottery Method. Systematic sampling technique was used to select the mothers having children aged less than 12 months, and they aimed to reach 430 mothers. They collected the data by face to face interviews, the questionnaire was developed by the researchers.

The data analyses was described in the methodology of the manuscript. They presented descriptive tables, and cross tables. They also conducted binary logistic regression analysis to find out the factors affecting prelacteal feeding.

The authors described the methodology of the research very detailed. The findings were presented well, but there are small recommendations for Table 1, Table 3 and Table 4.

It is appropriate to order the percentages of the variables from bigger to smaller in the frequency tables (such as Ethnicity in Table 1). (if it is not an ordinal variable)

It is better to present the variables in the tables systematic (Table 1; such as child, mothers, fathers, family variables in Table 1)

Only the percentage of true answers can be given in Table 3.

Only pre-lacteal feeding practices column can be given in Table 4, and the percentage is not well understood, you can add % in the parentheses under pre-lacteal feeing box. You can erase “yes” also.

The study was only conducted in Jinka Town, South Omo Zone, Ethiopia. Larger studies was published for the other countries. Also, the findings of the study was already known, and they were found in the literature. So, you can emphasize why this study was conducted in this specific region such as ethnicity, low educational level of the mothers, etc. Also, please this explanation in introduction and discussion sessions.

This study was only conducted in Jinka Town, South Omo Zone, Ethiopia, this can be accepted as a limitation. This is representative for only this region.

Reviewer #2: Overall comment:

This study aimed to determine the prevalence of pre-lacteal feeding practices and associated factors among children aged less than 12 months in Jinka Town, Ethiopia.

This is a very important topic. More detailed analysis and interpretation of the results are recommended in order to provide more useful recommendations to the Ministry of Health and others creating and delivering breastfeeding and infant and young child feeding messages.

External editing is recommended.

Specific comments:

Keywords:

Suggest using more precise keywords to enable visibility in searches.

Key messages:

Abstract:

1. Line 24: not just in developing countries – delete “in developing countries”.

2. Line 39: “positively associated” is accurate but suggest “factors associated with prelacteal feeding”.

Introduction:

3. Line 47: spell out acronyms first time e.g. World Health Organisation (WHO).

4. Lines 47-49: reword the sentence to active “WHO recommends ….”

Methods:

5. Lines 161-73 need editing.

6. Lines 170-71: I’m not sure if this method of checking collinearity is appropriate and looking at Table 4, it’s hard to believe that there was not collinearity in the adjusted models. Probably need to explain better how collinearity was checked and how models were constructed, e.g. clarify in/dependent variables and variables adjusted for.

Results:

7. Lines 213-15: were these problems self-diagnosed or diagnosed by health professionals?

8. Line 218: “purported” does not seem to be the best word. Need to explain what is meant by it’s use.

9. Table 3: was “don’t know” a response option? Or were these “true/false” responses?

10. Table 4: see comment on collinearity. Indicate p-values for unadjusted ORs.

Discussion:

11. Lines 277-86: explain why the age of the child might make a difference in prelacteal feeding – recall? communication campaign? Similarly for other comparisons – clarify possible reasons for differences.

12. Please provide rates of breastmilk substitute feeding. Breastmilk substitutes are important prelacteal feeds that don’t seem to have been considered in the study. This may explain the contradiction with the Egypt study (lines 294-97) – are more educated mothers giving breastmilk substitutes? Use of breastmilk substitutes should be considered, if possible. Was data collected on breastmilk substitutes? If not, can the DHS or other studies provide information?

13. The discussion is quite superficial. It should explore explanations in greater depth. For example, lines 306-13 – is anything known about the content of breastfeeding counselling in the different studies? Describe “traditional beliefs that favour prelacteal feeding” (lines 319-20).

conclusion and Recommendation

14. A deeper discussion/analysis (see above) would enable stronger and more detailed recommendations.

General comment:

15. Editing for punctuation, vocabulary selection, syntax, consistency (pre-lacteal or prelacteal?), clarity, conciseness is recommended.

16. Dispense with the word “practice” and just use “prelacteal feeding”.

Reviewer #3: General Comments

1. This was a community-based cross-sectional study on the prevalence of pre-lacteal feeding and associated factors children aged<12 months; PPS sampling for N=430 mothers.

2. This is an important study for the setting BUT it is not clear if it adds any additional knowledge to what is already in the literature, and what has been found in other parts of Ethiopia. The slight difference in estimates may just be due to sampling differences. The results (prevalence of pre-lacteal feeding, knowledge and reason for the practice etc.) indicate that targeted interventions are needed in this region. Breastfeeding counseling , and ANC visits have to be promoted and the authors recommend this.

3. If the paper is accepted some edits must be made including English copy-editing.

Specific comments:

1. Even though prevalence is around not much more than 10%, and the results would likely be similar, it would be interesting to see prevalence ratios instead of odds ratios.

2. Sample size calculation: it is not clear why the design effect was considered. Was the clustering of children by area (Kebeles)? Also, it seems like since the goal was to estimate prevalence, the authors should have calculated a required SS to estimate the prevalence with a particular precision. What does “95% certainty and maximum discrepancy of 5% between122 the sample size and the underlining population” mean?

3. What are the 3 breastfeeding knowledge questions? These should be listed in a table or appendix even if a reference is given.

4. Multicollinearity – there needs to be clarification of how this was done. Standard error or R-square?

5. In the decisions for pre-lacteal feeding section, what are the denominator of the listed percentages? Looks like they are for the N=53 who practiced pre-lacteal feeding. Throughout, the n (numerator and denominator) has to be clearly stated

6. The manuscripts needs English copy-editing

6. PLOS authors have the option to publish the peer review history of their article (what does this mean?). If published, this will include your full peer review and any attached files.

Reviewer #1: No

Reviewer #2: No

Reviewer #3: Yes: Bareng A.S. Nonyane, PhD MSc

---

## [Author Response · Author response to Decision Letter 0]

21 Mar 2020

Dear Editor and Reviewer 

Sincerest thanks for your response and reviewers comments on our manuscript. We have modified the paper in response to the extensive and insightful editor and reviewer comments. We would be glad to respond to any further questions and comments that you may have.

Responses

For Reviewer 1

• It is appropriate to order the percentage of the variables from bigger to smaller in the frequency table?

Of course many literatures don’t give attention but to get the reader’s attention we understand to present systematically and we have modified according to the comment

• You can emphasize why this study was conducted in this specific region such as ethnicity, low educational level of the mothers, etc. Also, please this explanation in introduction and discussion sessions. We amend it accordingly

• The other comments are modified accordingly

For Reviewer 2

• Key words…. Modified accordingly

• Line 24 modified according to the comment

• Line 39 modified according to the comment 

• Line 47 WHO, Modified by World Health Organisation (WHO).

• Line 47-49 WHO recommends Modified according to the comment

• Line 161-173 editing, Edited

• Line 170-71, 

• Multi co linearity test was carried out to see the correlation between all independent variables using collinearity statistics which is tolerance > 0.1 and variance inflation factor < 10. If two of the variables are highly correlated, then this may be the possible source of multicollinearity. Of course standard error is also used for checking multicollinearity, in our case standard error is also less than 4, and moreover, multicollinearity is a state of very high inter-correlations or inter-associations among the independent variables. It is therefore multicollinearity can also be detected with the help of tolerance and its reciprocal, called variance inflation factor (VIF).

• Line 213-15 were these problems self-diagnosed or diagnosed by health professionals? Diagnosed by health professionals

• Line 218: “purported” does not seem to be the best word. Need to explain what is meant by it’s use. 

It was to say there belief on the aim/advantage of pre-lacteal feeding, just we removed the word since we also understand it is not the appropriate term

• Table 3: was “don’t know” a response option? Or were these “true/false” responses?

 It was True/false response and modified in the table also since the questionnaire was prepared as true/false option

• Table 4: see comment on collinearity. Indicate p-values for unadjusted ORs

Thank you the comment, Pvalue was provided in the foot note

• Lines 277-86: explain why the age of the child might make a difference in prelacteal feeding – recall? communication campaign? Similarly for other comparisons – clarify possible reasons for differences.

Mothers having children <24 months may face difficulties to remember what they fed their child. Also, it is assumed that mothers with good educational status have a high chance for visiting health centers. It is modified accordingly

• Please provide rates of breastmilk substitute feeding. Breastmilk substitutes are important prelacteal feeds that don’t seem to have been considered in the study. This may explain the contradiction with the Egypt study (lines 294-97) 

The pre-lacteal feeds found in this study were Plain water (22 mothers), glucose water (3 mothers), cow milk (3 mothers), butter (13 mothers), Tenadam (11 mothers)

• Are more educated mothers giving breast milk substitutes? 

In our study mothers with good educational status were feeding their child breast milk as the odds of providing pre-lacteal feeding among mothers who were unable to read and write were nearly five folds higher than those who were secondary and above educational level, where as in Egypt study they even recommend that Further education of the mothers and health staff about adverse effects of PLF is required as they thought those well-educated are poor in practicing exclusive breast feeding based on their data.

• The discussion is quite superficial. It should explore explanations in greater depth. For example, lines 306-13 – is anything known about the content of breastfeeding counselling in the different studies? The contents are similar throughout the country but there is variation in practicing the counseling during ANC , delivery and Post natal period

• Describe “traditional beliefs that favour prelacteal feeding” (lines 319-20).

 Described according to the comment

For Reviewer 3

• Even though prevalence is around not much more than 10%, and the results would likely be similar, it would be interesting to see prevalence ratios instead of odds ratios.

The prevalence of pre-lacteal feeding practice in this study was 53 (12.6%)

• Sample size calculation: it is not clear why the design effect was considered. Was the clustering of children by area (Kebeles)? Yes there was clustering in the kebele (urban kebele and rural kebele) and we used design effect of 1.5 to consider the loss of effectiveness by the use of cluster sampling, instead of simple random sampling.

• Also, it seems like since the goal was to estimate prevalence, the authors should have calculated a required SS to estimate the prevalence with a particular precision. What does “95% certainty and maximum discrepancy of 5% between122 the sample size and the underlining population” mean? 

This was to mean the sample size was calculated by assuming 95% confidence level and a margin of error of 5%

• What are the 3 breastfeeding knowledge questions? These should be listed in a table or appendix even if a reference is given. 

 It is provided in the supplementary information with the tools

• Multicollinearity – there needs to be clarification of how this was done. Standard error or R-square? Multi-collinearity test was carried out to see the correlation between all independent variables using collinearity statistics which is tolerance > 0.1, variance inflation factor < 10 and standard error which was less than 4

• In the decisions for pre-lacteal feeding section, what are the denominators of the listed percentages? Looks like they are for the N=53 who practiced pre-lacteal feeding. Throughout, the n (numerator and denominator) has to be clearly stated

Thank you for the comment and yes it is from N=53 and modified accordingly

---

## [Editor Report · Decision Letter 1]

10 Aug 2020

PONE-D-19-26600R1

Pre-lacteal Feeding Practices and Associated Factors among Mothers of Children Aged Less Than 12 Months in Jinka Town, South Ethiopia, 2018/19

PLOS ONE

Dear Dr.Bekele Sorrie,

Thank you for submitting your manuscript to PLOS ONE. After careful consideration, we feel that it has merit but does not fully meet PLOS ONE’s publication criteria as it currently stands. Therefore, we invite you to submit a revised version of the manuscript that addresses the points raised during the review process.

Disclosure: I participated as a reviewer for the initial evaluation of this manuscript.

Decision: Major revision

The manuscript requires extensive copy-editing for language usage before it can be considered for publication.

There are also some specific areas that need attention – and some of these can be addressed by scientific copy-editing.

Abstract

Replace the last sentence of methods section with “Adjusted odds ratios, 95% confidence intervals and p-values are reported”Remove capital letters in mid-sentences in the results sectionCopy-edit for language usage throughout.

Methods and all other sections: copy-editing by a scientific editor is required to clarify the technical details in these sections

We require that you thoroughly copy-edit your manuscript for language usage, spelling, and grammar. If you do not know anyone who can help you do this, you may wish to consider employing a professional scientific editing service. 

Whilst you may use any professional scientific editing service of your choice, PLOS has partnered with both American Journal Experts (AJE) and Editage to provide discounted services to PLOS authors. Both organizations have experience helping authors meet PLOS guidelines and can provide language editing, translation, manuscript formatting, and figure formatting to ensure your manuscript meets our submission guidelines. To take advantage of our partnership with AJE, visit the AJE website (http://learn.aje.com/plos/) for a 15% discount off AJE services. To take advantage of our partnership with Editage, visit the Editage website (www.editage.com) and enter referral code PLOSEDIT for a 15% discount off Editage services. If the PLOS editorial team finds any language issues in text that either AJE or Editage has edited, the service provider will re-edit the text for free

We look forward to receiving your revised manuscript.

Kind regards,

Bareng A. S. Nonyane

Academic Editor

PLOS ONE

---

## [Author Response · Author response to Decision Letter 1]

15 Sep 2020

Dear Editor and Reviewer 

Sincerest thanks for your response and reviewers comments on our manuscript. We have modified the English language of this paper in response to the comments given. We would be glad to respond to any further questions and comments that you may have.

---

## [Editor Report · Decision Letter 2]

30 Sep 2020

Pre-lacteal Feeding Practices and Associated Factors among Mothers of Children Aged Less Than 12 Months in Jinka Town, South Ethiopia, 2018/19

PONE-D-19-26600R2

Dear Dr. Muluken Bekele Sorrie

We’re pleased to inform you that your manuscript has been judged scientifically suitable for publication and will be formally accepted for publication once it meets all outstanding technical requirements.

Kind regards,

Bareng A. S. Nonyane

Guest Editor

PLOS ONE
---

## [Editor Report · Acceptance letter]

2 Oct 2020

PONE-D-19-26600R2 

Pre-lacteal feeding practices and associated factors among mothers of children aged less than 12 months in Jinka Town, South Ethiopia, 2018/19 

Dear Dr. Sorrie:

I'm pleased to inform you that your manuscript has been deemed suitable for publication in PLOS ONE. Congratulations! Your manuscript is now with our production department. 

Kind regards, 

on behalf of

Dr Bareng A. S. Nonyane 

Guest Editor

PLOS ONE